# Physiological and Transcriptional Responses of Industrial Rapeseed (*Brassica napus*) Seedlings to Drought and Salinity Stress

**DOI:** 10.3390/ijms20225604

**Published:** 2019-11-09

**Authors:** Ji Wang, Jiao Jiao, Mengjia Zhou, Zeyang Jin, Yongjian Yu, Mingxiang Liang

**Affiliations:** 1Department of Ecology, College of Resources and Environmental Sciences, Nanjing Agricultural University, Nanjing 210095, China; 2016103004@njau.edu.cn (J.W.); 2016103003@njau.edu.cn (J.J.); 2017103004@njau.edu.cn (M.Z.); 2017103003@njau.edu.cn (Z.J.); 2015103003@njau.edu.cn (Y.Y.); 2Jiangsu Key Lab of Marine Biology, Nanjing 210095, China

**Keywords:** *Brassica napus*, drought, proline, salinity, transcriptomic analysis, transcription factors

## Abstract

Abiotic stress greatly inhibits crop growth and reduces yields. However, little is known about the transcriptomic changes that occur in the industrial oilseed crop, rapeseed (*Brassica napus*), in response to abiotic stress. In this study, we examined the physiological and transcriptional responses of rapeseed to drought (simulated by treatment with 15% (*w*/*v*) polyethylene glycol (PEG) 6000) and salinity (150 mM NaCl) stress. Proline contents in young seedlings greatly increased under both conditions after 3 h of treatment, whereas the levels of antioxidant enzymes remained unchanged. We assembled transcripts from the leaves and roots of rapeseed and performed BLASTN searches against the rapeseed genome database for the first time. Gene ontology analysis indicated that DEGs involved in catalytic activity, metabolic process, and response to stimulus were highly enriched. The Kyoto Encyclopedia of Genes and Genomes (KEGG) pathway analysis revealed that differentially expressed genes (DEGs) from the categories metabolic pathways and biosynthesis of secondary metabolites were highly enriched. We determined that myeloblastosis (MYB), *NAM*/*ATAF1-2*/*CUC2* (NAC), and APETALA2/ethylene-responsive element binding proteins (AP2-EREBP) transcription factors function as major switches that control downstream gene expression and that proline plays a role under short-term abiotic stress treatment due to increased expression of synthesis and decreased expression of degradation. Furthermore, many common genes function in the response to both types of stress in this rapeseed.

## 1. Introduction

Throughout their lifecycles, plants are subjected to various external environmental stresses including biotic stresses (such as weeds and diseases) and abiotic stresses (such as drought and salinity). In general, abiotic stress reduces crop yields by more than 50% compared to the less than 10% reduction caused by biotic stress [1]. Drought affects approximately 40% of the world’s agricultural land and is considered to be the most serious global agricultural problem [2]. Moreover, it is estimated that 800 million hectares of land have been salinized to some extent [3]. The limiting effect of drought and salinity on plant growth is expected to increase due to global climate change.

Plants adapt to short periods of abiotic stress through physiological regulation. Plant survival increases through osmotic adjustment or the removal of reactive oxygen species (ROS). Proline and sugars play important roles in osmotic adjustment, while superoxide dismutase (SOD), peroxidase (POD), and catalase (CAT) mediate ROS removal. In addition to physiological regulation, external stress signals are transduced in plants that induce stimulus-specific changes in gene expression. Plants have several common stress-related genes that allow them to withstand different adverse conditions [1,4].

Rapeseed (*Brassica napus*) is the third most important oilseed crop after soybean (*Glycine max*) and palm (*Trachycarpus sortunei*) [5]. Rapeseed cultivars with low glucosinolate and erucic acid contents are used to produce edible oils, animal feed, and biodiesel, whereas rapeseed cultivars with high erucic acid content have industrial applications. Since drought and salinity severely affect yield in rapeseed, changes in gene expression under different abiotic stress treatments have been studied to identify stress-specific genes for use in genetic engineering. For example, transcriptomic differences in two rapeseed varieties with different levels of drought resistance were identified at seven days after germination [6]. Differentially expressed genes (DEGs) were also identified in the roots of rapeseed seedlings under salinity stress soon after germination [7]. Transcriptomic and epigenomic comparisons have revealed that the expression levels of several drought-responsive genes in rapeseed seedlings depend on methylation patterns in the genome [8].

Transcriptome analysis is a powerful tool for identifying stress-related genes due to the ability to fully analyze gene transcription. Rapeseed (AACC genome) originated from a hybridization between *Brassica rapa* (AA genome) and *Brassica oleracea* (CC genome) [9]. However, even though sequencing of the rapeseed genome was completed in 2014 [10], most transcriptome analyses performed to date have been based on the NCBI non-redundant (NR) database [6], *B. rapa* genome database [7], or both the *B. rapa* and *B. oleracea* genomes [8]. Although rapeseed is closely related to other Brassica species, extensive genomic differences exist in the Brassicaceae family [10]. There are several reports of the transcriptome analysis of *B. napus* under different abiotic stresses such as freezing [11], silicon supply [12], metal stress [13], drought [14], and salt [15]. However, no common abiotic stress-related genes have been identified through the transcriptome analysis of rapeseed under both drought and salinity stress. After elaborating the mechanism behind abiotic stresses, new agronomic and breeding practices could be better applied to increase its stress adaption [16].

In the current study, we analyzed the transcriptome of an industrially important salinity-resistant rapeseed variety with high erucic acid content (Nanyanyou-1; resistant to 0.2–0.3% salinity) under drought (simulated by treatment with 15% (*w*/*v*) PEG 6000) and salinity (150 mM NaCl) stress. Our results indicated that proline metabolism played an important role in rapeseed under short-term abiotic stress (3 h). More DEGs were identified under drought vs. salinity stress and in the roots vs. leaf tissue. The findings of this study lay the foundation for developing industrially important rapeseed varieties with tolerance to drought and salinity stress using genetic engineering techniques and effective breeding programs.

## 2. Results

### 2.1. Physiological Characteristics of B. napus under Abiotic Stresses

Long-term drought and salinity stress significantly affect the physiological and biochemical functioning of plants. However, there were no obvious differences in superoxide dismutase (SOD), peroxidase (POD), catalase (CAT), H_2_O_2_, or soluble sugar contents in rapeseed seedlings after 3 h of treatment with 150 mM NaCl or 15% (*w*/*v*) PEG 6000 (Figure 1A–E). Malondialdehyde (MDA) and relative water contents were slightly different in the drought and salinity-treated seedlings when compared to the untreated control (Figure 1F,G), but no difference was detected between the two abiotic treatments. In contrast, both stress treatments triggered a marked increase in proline content (Figure 1H), which increased by more than 50% under NaCl treatment and was doubled under polyethylene glycol (PEG) treatment.

### 2.2. Transcriptomic Sequencing and Unigene Assembly

To further explore the molecular mechanism underlying the physiological responses of rapeseed to drought and salinity stress, the roots and leaves of stress-treated plants were collected, and 18 samples were used to construct transcriptome libraries and sequenced on the Illumina HiSeq platform. 

On average, approximately 6.64 Gb were generated per sample (Table 1). After mapping sequenced reads to the *B. napus* reference genome and reconstructing the transcripts using HISAT, an average of 66.27% mapped reads had met the requirements for further comparison. Ultimately, 42,251 transcripts were obtained from all samples including 4251 transcripts of novel genes that contain features not present in the reference annotation and 7127 long noncoding RNAs (Table 1). The value of Q30 exceeded 95%, suggesting that the sequencing data were reliable.

### 2.3. Analysis of Differentially Expressed Genes

DEGs were defined based on a fold change ≥ 2.00 and adjusted *p*-value ≤ 0.05. We identified 913 DEGs (502 upregulated and 411 downregulated) in CK-L (control, leaves) vs. D-L (drought, leaves); 3879 DEGs (2097 upregulated, 1782 downregulated) in CK-R (CK, roots) vs. D-R (drought, roots); 95 DEGs (53 upregulated, 42 downregulated) in CK-L vs. S-L (salinity, leaves); and 616 (190 upregulated, 426 downregulated) DEGs in CK-R vs. S-R (salinity, roots) (Appendix A, fold change ≥ 4.00 also listed here). There were more DEGs under drought treatment than under high-salinity treatment and many more DEGs in roots compared to leaves (Figure 2). More DEGs were upregulated in roots under drought stress than in roots under salinity stress, but more DEGs were downregulated in roots under salinity stress than in roots under drought stress, highlighting the different expression patterns of DEGs under these two abiotic stresses.

As illustrated in Venn diagrams (Figure 2), 102 DEGs were upregulated and 135 DEGs were downregulated in both leaves and roots under PEG treatment. In contrast, only four DEGs were upregulated and one DEG was downregulated in both leaves and roots under high-salinity treatment. No common DEGs were detected in all tissues under both treatments (Appendix A).

We performed hierarchical clustering analysis based on log_2_ fold change values to compare the expression patterns of upregulated and downregulated genes between groups (Appendix A). Basically, the transcriptional profiles between leaves and roots under salinity treatment were similar and also similar to that of roots under drought treatment to the extent. In both roots and leaves, DEGs in the drought-stressed group exhibited greater differences in expression relative to the untreated control than those in the salinity-treated group. In addition, most DEGs in roots showed greater differences in expression relative to the untreated control group than that of DEGs in leaves under both treatments.

### 2.4. GO and KEGG Pathway Enrichment Analysis

Gene ontology (GO) is a common functional classification system used to identify genes with specific functions in three categories: cellular component, molecular function, and biology process. Ultimately, 812 out of 913 DEGs in drought-stressed leaves, 3366 out of 3879 DEGs in drought-stressed roots, 82 out of 95 DEGs in salinity-stressed leaves, and 524 out of 616 DEGs in salinity-stressed roots were assigned to GO categories (Figure 3). Most genes in each category were assigned to one of three sub-categories (cell part, cell, and organelle from cellular component; catalytic activity, binding, and transporter activity from molecular function; metabolic process, cellular process, and response to stimulus from biological process).

To conduct a comprehensive pathway analysis of the DEGs in each group, we performed a Kyoto Encyclopedia of Genes and Genomes (KEGG) pathway analysis. Due to the limitations of the KEGG database, only 80.7% of the DEGs were assigned to KEGG pathways. KEGG pathways were assigned based on a threshold value of *p* ≤ 0.02. There were 24 significantly enriched pathways for DEGs in CK-L vs. D-L and 39, 3, and 16 pathways for CK-L vs. D-R, CK-R vs. S-L, and CK-L vs. S-R, respectively. Under PEG treatment, DEGs involved in metabolic pathways and biosynthesis of secondary metabolites were highly enriched (Table 2A,B). Under NaCl treatment, the same pathways were enriched in roots as those enriched in roots under drought treatment, while the cutin, suberin, and wax biosynthesis pathways and ABC transporters were markedly enriched in leaves under salinity treatment (Table 2C,D).

### 2.5. Analysis of Differentially Expressed Transcription Factors

To investigate the importance of transcription factors (TFs) in the stress resistance regulatory network, we analyzed differentially expressed TF genes. We identified 78, 587, 3, and 39 TF genes in CK-L vs. D-L, CK-R vs. D-R, CK-L vs. S-L, and CK-R vs. S-R, respectively. Both the numbers and classifications of TF genes differed in roots and leaves under both treatments (Figure 4). Under PEG treatment, the basic region-leucine zipper (bZIP), myeloblastosis (MYB), plant AT-rich sequence- and zinc-binding (PLATZ), and basic/helix-loop-helix (bHLH) TF family genes were enriched in leaves, while the MYB, NAC, bHLH, APETALA2/ethylene-responsive element binding proteins (AP2-EREBP), and WRKY domain contained proteins (WRKY) TF family genes were enriched in roots. Under NaCl treatment, AP2-EREBP and C2C2-CO-like TF genes were enriched in leaves, while *NAM*/*ATAF1-2*/*CUC2 (*NAC), MYB, WRKY, and G2-like TF genes were enriched in roots. Moreover, 435 out of 665 TF genes were upregulated under PEG treatment, whereas similar numbers of TF genes were up- or downregulated under high-salinity treatment (Appendix A).

As far as proline metabolism, the promoters of DEGs encoding *BnOAT* (BnaA06g36140D), *BnP5CS1* (BnaA03g18760D, BnaA05g05760D, BnaC03g72600D, BnaC04g05620D), and *BnP5CS2* (BnaC04g55570D, BnaA04g03460D) were further analyzed (Appendix A). The analysis of putative cis-regulatory elements showed that all of these DEGs harbored binding sites for RAV, NAC, bZIP, and MYB. Except for BnaA06g36140D, the other members harbored binding sites for WRKY. Binding sites for AP2-ERF only existed in the promoters of DEGs encoding *BnP5CS1*.

### 2.6. Validation of Differentially Expressed Genes by Quantitative RT-PCR Analysis

To further verify the reliability of the sequencing data, we randomly selected 10 DEGs (five under PEG treatment, five under NaCl treatment) and subjected them to qRT-PCR analysis. Although the fold change values of these DEGs were not the same as those obtained by sequencing, the trends were consistent (Figure 5 and Appendix A). 

## 3. Discussion

### 3.1. The Important Role of Proline in Abiotic Stress Response

Proline is an osmotic regulatory substance with a low molecular weight, high water solubility, and no electrostatic charge at physiological pH [17,18]. Various plant species such as *Nicotiana tabacum* [19], *Sesamum indicum* [20], *Medicago truncatula* [21], *Oryza sativa* [22], *Eremochloa ophiuroides* [23], *Brassica rapa ssp. Pekinensis* [24], and *Arabidopsis thaliana* [25] accumulate proline during adaptation to osmotic stress.

In *Brassica napus*, proline levels sharply increase under both drought [26] and high salinity stress [27]. Indeed, in the current study, both drought and salinity stress led to the significant accumulation of proline in rapeseed (Figure 1H). Therefore, we further analyzed DEGs involved in proline metabolism. We detected two DEGs that were upregulated in leaves under PEG treatment, five DEGs (one upregulated and four downregulated) in roots under drought treatment, and six DEGs (three upregulated and three downregulated) in roots under NaCl treatment (Appendix A).

Both the biosynthesis and degradation of proline play important roles in proline accumulation. Pathways involved in proline biosynthesis include the glutamate (Glu) and ornithine (Orn) pathways. In the Glu pathway, glutamate is converted to GSA (glutamate-γ-semialdehyde) via a reaction catalyzed by P5CS (Δ^1^-pyrroline-5-carboxylate synthetase). GSA is further reduced to proline by P5CR (Δ^1^-pyrroline-5-carboxylate reductase). When proline accumulates to a certain level, the expression of *P5CS* is suppressed, while that of *ProDH* is induced, leading to proline degradation. In the Orn pathway, ornithine loses its δ-amino group to generate GSA through transamination, a process mediated by OAT (ornithine aminotransferase). GSA participates in the Glu pathway to generate proline. Therefore, the key enzymes in the proline biosynthesis pathway are P5CS, P5CR, and δ-OAT, whereas the key enzyme in the proline degradation pathway is ProDH (proline dehydrogenase) [28,29].

In the current study, only DEGs in the Glu pathway were identified in CK-R vs. S-R and CK-R vs. D-R, whereas DEGs in both pathways were induced in CK-L vs. D-L. No DEGs in the Glu or Orn pathway were enriched in CK-L vs. S-L. Genes encoding BnP5CS proteins in the Glu pathway were upregulated under stress treatment including BnP5CS1 (BnaA03g18760D, BnaA05g05760D, BnaC03g72600D, BnaC04g05620D), and BnP5CS2 (BnaC04g55570D, BnaA04g03460D) (Figure 6). Similarly, BnaA06g36140D, a gene encoding BnOAT in the Orn pathway, was also upregulated in leaves under drought stress. Furthermore, we identified genes encoding ProDH that were downregulated in roots under both treatments including BnaC02g38230D, BnaA06g39660D, and BnaAnng07910D (Appendix A). Thus, when treated with PEG or NaCl, rapeseed seedlings synthesize more proline by increasing the expression of *P5CS* or *OAT*, and they also reduce the degradation of proline by suppressing *ProDH* expression.

In *Arabidopsis thaliana*, which is a close relative of *B. napus*, P5CS is encoded by two similar regulatory genes named *AtP5CS1* and *AtP5CS2*, while ProDH is encoded by *AtPDH*. When treated with ABA or NaCl, Arabidopsis plants accumulate increased levels of proline due to high *AtP5CS1* expression and low *AtPDH* expression [30]. When treated with exogenous H_2_O_2_, proline accumulation increases in plants, which is associated with the activation of the Glu and Orn proline biosynthesis pathways [31]. *AtP5CS1* and *AtP5CS2* are induced by cold, NaCl, abscisic acid (ABA), desiccation, light, heat, rehydration, and brassinolide treatment [32]. In *B. napus*, proline accumulation during priming and post-priming germination is associated with the strong upregulation of *P5CSA* and the downregulation of *PDH* [33].

The proline pathway is regulated by several types of TFs. In Arabidopsis, *AtP5CS1/2*, *AtP5CR*, and *AtOAT* contain several TF binding sites 1000 bp upstream of their promoter regions that bind TFs such as MYB, bZIP, AP2/EREBP, WRKY, and RAV TFs [32]. In rice, genes in the Glu pathway are thought to be targeted by many TFs. For example, 24 different classes of TFs have binding sites in the promoters of *OsP5CS1/2* and *OsP5CR* [34]. In *Medicago truncatula*, *MtMYBS1*, a MYB TF gene, is induced by NaCl, PEG, and ABA treatment, and MtMYBS1 enhances the transcription of *P5CS* [35]. Transgenic *Betula platyphylla* plants overexpressing *BplMYB46* exhibited improved salinity and osmotic tolerance due to the increased expression of *P5CS* genes [36]. In addition, AP2/EREBP upregulates *AtP5CS1* in response to low water potential [37]. In rice, heterologous expression of *JERF1*, which encodes a tomato ERF protein, increased the accumulation of proline by upregulating *OsP5CS* [38]. Of course, even in the same TF family, different members could function differently. The analysis of phylogenetic trees showed that reported MtMYBS1 and BplMYB46 had similarities with BnMYBs including BnaA01g32800D and BnaA03g29470D, and BnaA05g30870D. JERF1 had similarities with BnaA07g30130D (Appendix A). This implies that these TFs may have similar function in the regulation of expression of proline biosynthetic genes. In the current study, we also identified several binding sites for TFs such as MYB, WRKY, and bZIP in the 1000-bp upstream regions of the *BnP5CS* and *BnOAT* promoters (Appendix A). The identified promoter region all had binding sites of MYB and most of them also had AP2-ERF binding sites (Appendix A). Perhaps in response to abiotic stress, *BnP5CSs* and *BnOAT* are upregulated by those TFs in rapeseed to induce the accumulation of proline. Without doubt, the region far away those upstream 1000-bp or 3′ untranslated region may have binding sites that also regulate the transcription of proline-related genes. 

### 3.2. The Multiple Transcripts in Response to Salinity and Drought Stresses

Based on transcriptome analysis, more DEGs were present in the roots than in leaves after 3 h of abiotic stress treatment (Figure 2 and Appendix A); this result is consistent with previous findings [1]. Using microarray analysis, 624 DEGs were detected in Arabidopsis roots compared to only 285 in leaves in response to a 3-h salinity treatment. Compared to salinity treatment, more DEGs were detected in rapeseed following PEG treatment in the current study. Transcriptome analysis revealed ~1700 DEGs in rapeseed after 3 h vs. 24 h of salinity treatment, most of which were downregulated in the whole roots [7]. These findings indicate that DEG responses to abiotic stress differ depending on the plant species, treatment, and growth stage.

Based on KEGG analysis, the most important DEGs in both roots and leaves under drought stress and in roots under salinity stress are involved in the ‘metabolic pathway’ and ‘biosynthesis of secondary metabolites’ (Table 2A,B,D). It is not surprising that many DEGs are involved in secondary metabolite biosynthesis due to the important roles of these compounds in plants subjected to stress [39,40,41]. Several ABC transporter genes were differentially expressed in leaves under salinity stress (Table 2C). ABC transporters (ATP-binding cassette transporters) transport organic materials, especially secondary metabolites, for plant development [42]. ABC transporters also play important roles in cutin and wax formation [43]. Several DEGs related to cutin, suberin, and wax biosynthesis were identified in leaves under salinity stress, confirming the importance of secondary metabolites in abiotic stress responses.

MYB and MYB-related TF genes were highly upregulated under both salinity and drought stress (Figure 4). MYBs, which are characterized by a highly conserved DNA-binding MYB domain are present in all eukaryotes [44]. MYBs form a large gene family involved in development, metabolism, and stress responses [45,46]. MYBs also form homo- and heterodimers with other proteins to regulate various processes in plants [47].

In addition to MYB TF genes, NAC TF genes were highly upregulated by abiotic stress in rapeseed, especially salinity stress (Figure 4B). Like MYB TFs, NAC TFs form a large family, but are specific to plants [48]. In general, the N-termini of NAC TFs are highly conserved regions that function in DNA binding, and protein–protein interactions. Increasing evidence suggests that NAC TFs are involved in regulating abiotic or biotic stress responses [49].

AP2/EREBP TFs contain DNA-binding AP2 domains, which are also unique to plants [50]. This large TF family participates in diverse stress responses [51]. Plants use abscisic acid (ABA)-dependent and ABA-independent pathways to cope with abiotic stress. MYB TFs function in ABA-dependent pathways, whereas AP2/EREBP TFs function in ABA-independent pathways. Many AP2/EREBP TFs such as CBFs and DREBs have been extensively characterized and manipulated to increase abiotic stress resistance in plants [52].

Numerous DEGs identified in the current study showed the same expression patterns under both abiotic stress treatments (Appendix A). There were more DEGs shared in roots than in leaves under the two treatments. Most of these genes were categorized as encoding hypothetical proteins. Several genes involved in proline metabolism or abiotic stress pathways were induced in both the leaves and roots under abiotic stress, whereas those involved in primary metabolism tended to be downregulated under these treatments. These findings promote our understanding of the regulatory mechanisms underlying the drought and salinity stress responses in *B. napus* and lay the foundation for breeding cultivars with improved tolerance to abiotic stress.

## 4. Materials and Methods 

### 4.1. Plant Materials and Growth Conditions

*Brassica napus* ‘Nanyanyou-1’ seeds were harvested in Nanjing, Jiangsu Province, China (31~32°N, 118~119°E) in 2016 and kept at Nanjing Agricultural University. The seeds were first germinated on wet gauze (soaked with water) in the incubator with a light intensity of 392~415 μmol m^−2^s^−1^ during a daily cycle consisting of 16 h of light at 25 °C and 8 h of darkness at 18 °C. The seedlings were then transferred into a ½ Hoagland nutrient solution to conduct a hydroponic experiment under the same culture conditions for nearly 20 days until the fourth leaves had extended. The plants were treated with ½ Hoagland nutrient solution containing 15% (*w*/*v*) PEG 6000 or 150 mM NaCl for drought and salinity stress treatments, respectively; seedlings treated with ½ Hoagland nutrient solution alone were used as the control. Each treatment included three biological replications. Leaves and roots were harvested individually after 3 h of treatment, immediately frozen in liquid nitrogen, and stored at 80 °C until use for RNA extraction and physiological and biochemical analyses.

### 4.2. Physiological and Biochemical Analyses

For water content (WC) determination, the fresh weight (FW) of the whole seedlings was first measured and then dried to a constant weight at 65 °C for 72 h to obtain the dry weight (DW). WC was calculated as follows: WC (%) = (FW − DW)/FW × 100%. MDA (malondialdehyde), Pro (proline), SOD (superoxide dismutase), POD (peroxidase), CAT (catalase), H_2_O_2_, and soluble sugar contents were determined using microdetermination kits (Suzhou Comin Biotechnology Co. Ltd, Suzhou, China). One-way ANOVA using the Least Significant Difference (LSD) method was conducted with SPSS 19.0 software (SPSS Corp., Chicago, IL, USA) to evaluate significant differences among treatments. Figures were drawn using SigmaPlot 10 (Systat Software, Inc., Berlin, Germany) software.

### 4.3. RNA Extraction, cDNA Library Construction, and Sequencing

Eighteen rapeseed plant samples including roots and leaves were sent to Beijing Genomics Institute (BGI) for RNA extraction and cDNA library construction. Total RNA was extracted using the TRIzol reagent (Invitrogen, Carlsbad, CA, USA) for each biological replicate and was treated with DNase. The mRNA were purified from total RNA using magnetic Oligo (dT) beads and then fragmented by the fragmentation buffer. The cDNA was synthesized using the mRNA fragments as templates and the library was built. Agilent 2100 Bioanaylzer and ABI StepOnePlus Real-Time PCR System were used in the quantification and qualification of the sample library. Next, the library was sequenced using Illumina HiSeq 4000 to generate 150 bp, PE type data. The RNA-seq data was uploaded to the NCBI with the accession number PRJNA579479.

### 4.4. Genome Mapping and Quantification of Gene Expression

HISAT (version: v0.1.6-beta, Maryland, USA. Parameters: –phred64 –sensitive –no-discordant –no-mixed -I 1 -X 1000) was used for genome mapping [53]. StringTie (version: v1.0.4, parameters: -f 0.3 -j 3 -c 5 -g 100 -s 10000 -p 8) was used to reconstruct transcripts, and Cuffcompare (version: v2.2.1, Massachusetts, USA. Parameters: -p 12) was used to identify novel transcripts in our samples based on genome annotation information [54]. After novel transcript detection, novel coding transcripts were merged with the reference transcript (CoGe, https://genomevolution.org/CoGe/) to obtain a complete reference, and Bowtie2 (version: v2.2.5, Maryland, USA. Parameters: -q –phred64 –sensitive –dpad 0 –gbar 99999999 –mp 1, 1 –np 1 –score-min L, 0, -0.1 -I 1 -X 1000 –no-mixed –no-discordant -p 1 -k 200) was used to map clean reads [55]. Gene expression levels were also calculated for each sample using RSEM (version: v1.2.12, WI, USA. Parameters: default) [56].

### 4.5. Identification and Functional Enrichment Analysis of DEGs

DEseq2 (version: v3.10, Berlin, Germany.) was used to identify DEGs based on the parameters of an adjusted fold change of ≥2.00 and *p*-value ≤ 0.05 [57]. DEG lists were uploaded and analyzed online, and Venn diagrams were constructed (http://bioinformatics.psb.ugent.be/webtools/Venn/). Hierarchical clustering of the DEGs was performed using MeV-4.9.0 software (http://sourceforge.net/projects/mev-tm4/). The GO (Gene Ontology) annotation results were visualized, compared, and plotted using the online program WEGO (http://wego.genomics.org.cn) [58]. We also performed KEGG (Kyoto Encyclopedia of Genes and Genomes) pathway classification for functional enrichment of the DEGs [59]. For p-value correction (to control the false discovery rate, FDR), the rigorous Bonferroni correction method was used [60]. DEGs with a corrected p-value and FDR of ≤ 0.001 were defined as significantly enriched for both types of functional enrichment analyses.

### 4.6. Prediction of DEGs Encoding Transcription Factors

The open reading frame (ORF) of each DEG was identified using getorf (http://genome.csdb.cn/cgi-bin/emboss/help/getorf) [61]. The ORFs were aligned to transcription factor (TF) domains using HMMsearch (http://hmmer.org) [62] and TFs were identified as described in PlantfDB (http://planttfdb.cbi.pku.edu.cn/). The overall distribution of TFs in the two organs under both treatments were respectively counted and analyzed.

### 4.7. Analysis of Putative Cis-Regulatory Elements Involved in Proline Metabolism

The upstream 1000 bp genomic sequence of *BnP5CS* and *BnOAT* before the ATG codon were subjected to cis-element analysis. The sequences were analyzed using PLACE (Plant Cis-element Regulatory DNA Elements, http://www.dna.affrc.go.jp/PLACE/) and plantCARE (plant cis-acting regulatory element, http://bioinformatics.psb.ugent.be/webtools/plantcare/html/).

### 4.8. Verification of Differential Gene Expression by Quantitative Reverse-Transcription PCR

The transcript levels of genes expressed in different tissues or under different treatments were quantified by quantitative reverse-transcription PCR (qRT-PCR) using the same samples chosen for transcriptome analysis. The experiments were performed in an Applied Biosystems 7500 real-time PCR system using a SYBR Premix ExTaq Kit (TaKaRa Code: DRR041A, Japan). The reaction system and procedure used were described previously [63]. Data were processed using the 2^−ΔΔCT^ method. The primers are listed in Appendix A.

## 5. Conclusions

In this study, we examined the physiological and transcriptional response of industrial rapeseed to drought and salinity treatment. Short-term abiotic treatments caused a significant increase in proline content in the seedlings, while the levels of antioxidant enzymes remained unchanged. GO and KEGG analysis indicated that most of the identified DEGs were involved in the metabolic process, response to stimulus, or biosynthesis of secondary metabolites. Some stress-responsive DEGs were shared between plants subjected to drought and salinity treatments.

## Figures and Tables

**Figure 1 ijms-20-05604-f001:**
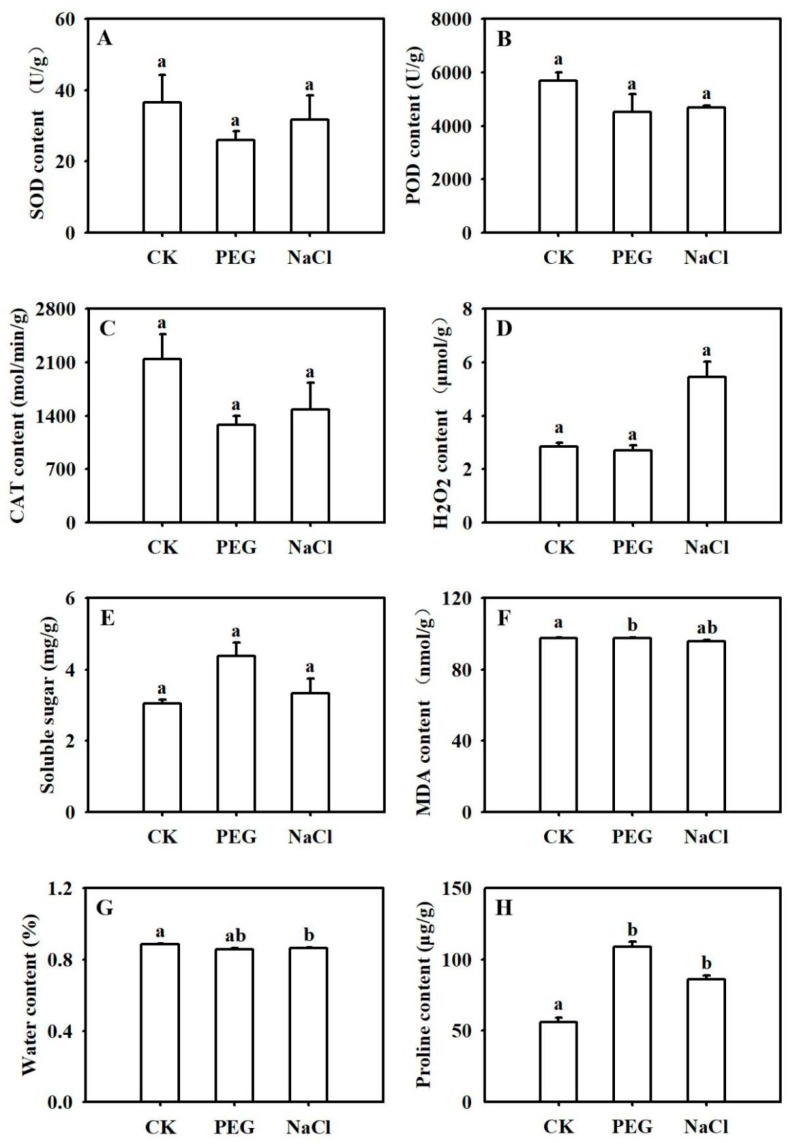
Effects of drought and salinity stress on physiological and biochemical parameters in *B. napus*. Superoxide dismutase—SOD (**A**), peroxidase—POD (**B**), catalase—CAT (**C**), H_2_O_2_ (**D**), soluble sugar (**E**), malondialdehyde—MDA (**F**), water (**G**), and proline—PRO (**H**) contents were determined after 3 h of drought (simulated by polyethylene glycol (PEG)) or salinity (NaCl) treatment. CK represents the control group, (i.e., seedlings treated with ½ Hoagland nutrient solution); PEG represents seedlings treated with 15% (*w*/*v*) PEG 6000 plus ½ Hoagland nutrient solution for 3 h. NaCl represents seedlings treated with 150 mM NaCl plus ½ Hoagland nutrient solution for 3 h. Each data point represents the mean of three samples ± SE. Columns with different letters in each graph indicate significant differences based on Duncan’s multiple range tests at *p* < 0.05 among treatments.

**Figure 2 ijms-20-05604-f002:**
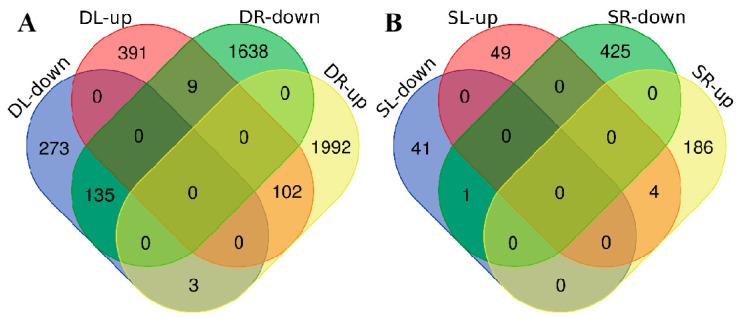
Venn diagrams of differentially expressed genes (DEGs). (**A**) DEGs between the control (CK) and drought (simulated by PEG) treatment. DL-down/DR-down indicate downregulated DEGs in leaves/roots under PEG treatment compared to the control. DL-up/DR-up indicate upregulated DEGs in leaves/roots under PEG treatment compared to the control. (**B**) SL-down/SR-down indicate downregulated DEGs under NaCl treatment compared to the control. SL-up/SR-up indicate upregulated DEGs under NaCl treatment in leaves/roots compared to the control.

**Figure 3 ijms-20-05604-f003:**
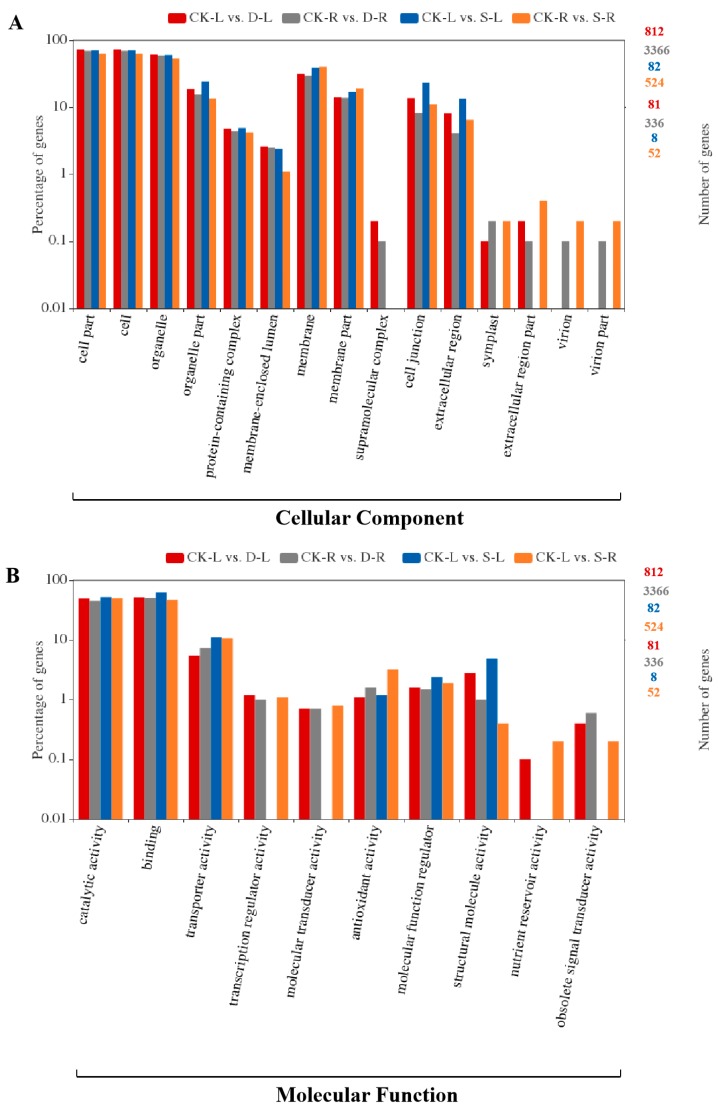
Gene ontology (GO) functional classification of DEGs from four treatment groups in *B. napus*. A total of 4784 DEGs were classified into GO terms from three ontologies involving cellular components (**A**), molecular function (**B**), and biological processes (**C**). CK-L (CK, leaves) vs. D-L (drought, leaves), CK-R (CK, roots) vs. D-R (drought, roots), CK-L vs. S-L (salinity, leaves), CK-R vs. S-R (salinity, roots) represent DEGs under these two abiotic stresses. The y-axis on the right indicates the number of genes in each category. The y-axis on the left indicates the percentage of specific genes in each category.

**Figure 4 ijms-20-05604-f004:**
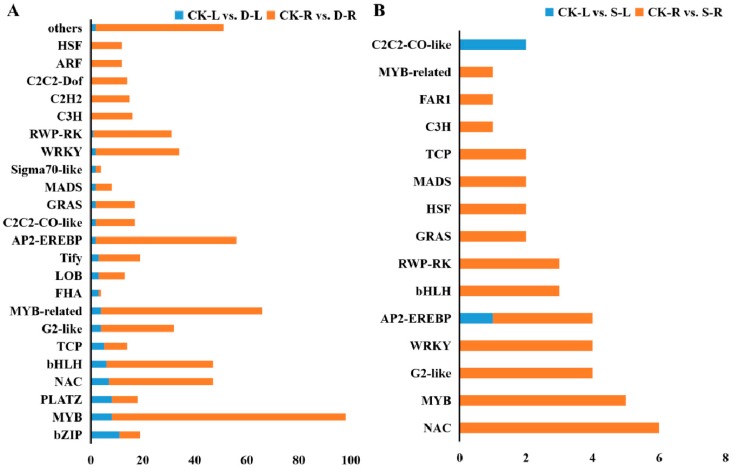
Differentially expressed transcription factors under drought or salinity stress in *B. napus*. Representative DEGs under (**A**) drought stress and (**B**) salinity stress, respectively. CK-L (CK, leaves) vs. D-L (drought, leaves), CK-R (CK, roots) vs. D-R (drought, roots), CK-L vs. S-L (salinity, leaves), CK-R vs. S-R (salinity, roots) represent DEGs under these two abiotic stresses.

**Figure 5 ijms-20-05604-f005:**
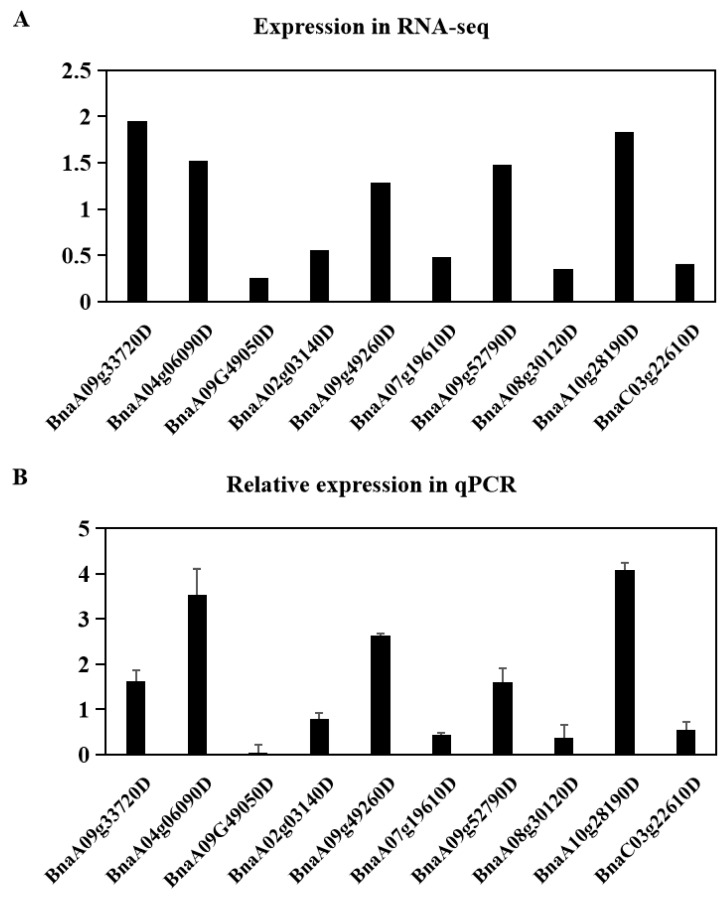
Relative expression levels of DEGs under drought (simulated by PEG) or salinity stress. Three-week-old rapeseed plants were treated with 15% (*w*/*v*) PEG 6000 or 150 mM NaCl for 3 h. (**A**) The upper panel showed the expression levels of the selected DEGs in RNA-seq data using their fold change value. (**B**) Total RNA was extracted from the leaves and roots for quantitative PCR (qRT-PCR) analysis. DEG transcript levels were normalized to that of the housekeeping gene *β-actin* before being compared to the control levels. BnaA09g33720D is a DEG in leaves under PEG treatment, whereas BnaA04g06090D, BnaA09G49050D, BnaA02g03140D, and BnaA09g49260D are DEGs in roots under PEG treatment. BnaA07g19610D and BnaA09g52790D are DEGs in leaves under salinity treatment, whereas BnaA08g30120D, BnaA10g28190D, and BnaC03g22610D are DEGs in roots under salinity treatment. Values represent the means and standard errors (SEs) of three biological samples. Each sample was analyzed by PCR in triplicate. The average expression level of each gene under the control treatment was set to 1.0.

**Figure 6 ijms-20-05604-f006:**
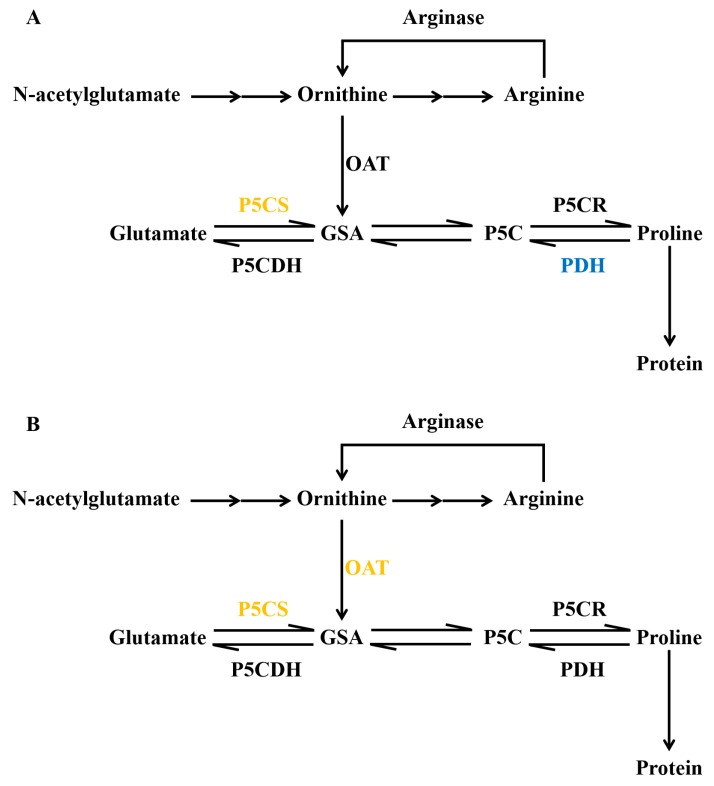
Major DEGs involved in proline metabolism in *B. napus*. (**A**) Proline metabolism pathway DEGs in CK-R (CK, roots) vs. D-R (drought, roots) and CK-R vs. S-R (salinity, roots). (**B**) Proline metabolism pathway DEGs in CK-L (CK, leaves) vs. D-L (drought, leaves). P5CS (Δ^1^-pyrroline-5-carboxylate synthetase), P5CDH (pyrrolidine-5-carboxylic acid dehydrogenase), GSA (glutamate-γ-semialdehyde), P5C (pyrroline-5-carboxylic acid), P5CR (Δ^1^-pyrroline-5-carboxylate reductase), PDH (proline dehydrogenase), and OAT (ornithine aminotransferase) function in proline metabolism. Yellow indicates that the underlying DEGs were upregulated, while blue indicates that the underlying DEGs were downregulated.

**Table 1 ijms-20-05604-t001:** Summary of sequencing reads after filtering and genome mapping.

Sample	Total Clean Reads (Mb)	Total Clean Bases (Gb)	Clean Reads Q30 (%)	Total Mapping Ratio
CK-L-1	44.93	6.74	95.03	60.59%
CK-L-2	44.07	6.61	96.06	64.09%
CK-L-3	45	6.75	95.94	64.47%
CK-R-1	44.27	6.64	95.34	62.21%
CK-R-2	44.42	6.66	95.81	59.86%
CK-R-3	44.06	6.61	95.94	60.16%
D-L-1	44.73	6.71	95.02	63.08%
D-L-2	45.13	6.77	95.1	63.50%
D-L-3	44.8	6.72	95.1	63.80%
D-R-1	44.97	6.74	95.4	60.51%
D-R-2	44.71	6.71	95.06	61.52%
D-R-3	41.5	6.23	95	62.03%
S-L-1	42.78	6.42	95.18	63.53%
S-L-2	44.42	6.66	95.18	62.31%
S-L-3	44.99	6.75	95.2	61.65%
S-R-1	44.9	6.73	95.25	61.69%
S-R-2	44.91	6.74	95.63	61.21%
S-R-3	42.11	6.32	95.09	60.84%
Total number of novel transcripts			42,251	
Coding transcripts			35,124	
Noncoding transcripts			7127	
Novel genes			4251	

Q30 indicates a quality score of 30, a 0.1% chance of error, and 99.9% confidence.

**Table 2 ijms-20-05604-t002:** KEGG pathways of DEGs.

**2A. KEGG pathways of DEGs in CK-L vs. D-L**				
#	**Pathway**	**Annotation (713)**	***p*-Value**	**Pathway ID**
1	**Metabolic pathways**	**210 (29.45%)**	**1.487446e-06**	**ko01100**
2	**Biosynthesis of secondary metabolites**	**133 (18.65%)**	**2.375184e-06**	**ko01110**
3	Pyruvate metabolism	20 (2.81%)	0.0003487765	ko00620
4	Arginine and proline metabolism	13 (1.82%)	0.001129843	ko00330
5	Glycine, serine and threonine metabolism	16 (2.24%)	0.001321938	ko00260
6	Glutathione metabolism	13 (1.82%)	0.001345293	ko00480
7	Fatty acid degradation	10 (1.4%)	0.001462133	ko00071
8	Glycerolipid metabolism	20 (2.81%)	0.001519717	ko00561
9	Indole alkaloid biosynthesis	6 (0.84%)	0.001620922	ko00901
10	Regulation of autophagy	13 (1.82%)	0.001845656	ko04140
11	Ribosome	37 (5.19%)	0.002576793	ko03010
12	Tryptophan metabolism	12 (1.68%)	0.003130347	ko00380
13	Caffeine metabolism	2 (0.28%)	0.003729469	ko00232
14	Ascorbate and aldarate metabolism	12 (1.68%)	0.004263276	ko00053
15	Histidine metabolism	7 (0.98%)	0.004263308	ko00340
16	mRNA surveillance pathway	27 (3.79%)	0.005933953	ko03015
17	alpha-Linolenic acid metabolism	9 (1.26%)	0.007378807	ko00592
18	Ubiquinone and other terpenoid-quinone biosynthesis	9 (1.26%)	0.008113964	ko00130
19	Cysteine and methionine metabolism	15 (2.1%)	0.01011539	ko00270
20	Steroid biosynthesis	7 (0.98%)	0.01100777	ko00100
21	Pentose and glucuronate interconversions	20 (2.81%)	0.01620864	ko00040
22	Lysine degradation	8 (1.12%)	0.01643623	ko00310
23	Galactose metabolism	13 (1.82%)	0.019456	ko00052
24	Carotenoid biosynthesis	8 (1.12%)	0.0198106	ko00906
**2B. KEGG pathways of DEGs in CK-R vs. D-R**				
#	**Pathway**	**Annotation (3034)**	***p*-Value**	**Pathway ID**
1	**Metabolic pathways**	**926 (30.52%)**	**1.348677e-29**	**ko01100**
2	**Biosynthesis of secondary metabolites**	**570 (18.79%)**	**2.264539e-23**	**ko01110**
3	Photosynthesis–antenna proteins	26 (0.86%)	1.749018e-15	ko00196
4	Phenylpropanoid biosynthesis	132 (4.35%)	6.312239e-15	ko00940
5	Glycolysis/Gluconeogenesis	105 (3.46%)	9.470061e-12	ko00010
6	Carbon metabolism	151 (4.98%)	1.339665e-11	ko01200
7	Pyruvate metabolism	70 (2.31%)	7.136255e-08	ko00620
8	Carbon fixation in photosynthetic organisms	60 (1.98%)	1.608047e-07	ko00710
9	Photosynthesis	37 (1.22%)	4.541068e-07	ko00195
10	Plant hormone signal transduction	172 (5.67%)	6.098387e-07	ko04075
11	Arginine and proline metabolism	43 (1.42%)	3.276459e-06	ko00330
12	Circadian rhythm–plant	66 (2.18%)	4.310124e-06	ko04712
13	Nitrogen metabolism	35 (1.15%)	2.766522e-05	ko00910
14	Alanine, aspartate and glutamate metabolism	43 (1.42%)	3.614134e-05	ko00250
15	Sulfur metabolism	25 (0.82%)	0.0001004235	ko00920
16	Valine, leucine and isoleucine degradation	32 (1.05%)	0.0001541904	ko00280
17	Fructose and mannose metabolism	44 (1.45%)	0.0002872131	ko00051
18	Glycine, serine and threonine metabolism	46 (1.52%)	0.001046218	ko00260
19	Inositol phosphate metabolism	34 (1.12%)	0.001077126	ko00562
20	Stilbenoid, diarylheptanoid and gingerol biosynthesis	45 (1.48%)	0.001329768	ko00945
21	Biosynthesis of unsaturated fatty acids	19 (0.63%)	0.001333774	ko01040
22	Galactose metabolism	45 (1.48%)	0.002157363	ko00052
23	Starch and sucrose metabolism	131 (4.32%)	0.002175542	ko00500
24	Flavonoid biosynthesis	31 (1.02%)	0.002204898	ko00941
25	Taurine and hypotaurine metabolism	14 (0.46%)	0.002282244	ko00430
26	Zeatin biosynthesis	18 (0.59%)	0.003346513	ko00908
27	Citrate cycle (TCA cycle)	33 (1.09%)	0.00335235	ko00020
28	Diterpenoid biosynthesis	25 (0.82%)	0.003665961	ko00904
29	Pentose phosphate pathway	44 (1.45%)	0.004369615	ko00030
30	Biosynthesis of amino acids	113 (3.72%)	0.004736771	ko01230
31	beta-Alanine metabolism	27 (0.89%)	0.005233566	ko00410
32	Flavone and flavonol biosynthesis	16 (0.53%)	0.006075288	ko00944
33	Limonene and pinene degradation	40 (1.32%)	0.009214323	ko00903
34	Tyrosine metabolism	25 (0.82%)	0.01085029	ko00350
35	Glyoxylate and dicarboxylate metabolism	44 (1.45%)	0.01298711	ko00630
36	Amino sugar and nucleotide sugar metabolism	60 (1.98%)	0.0143173	ko00520
37	Cysteine and methionine metabolism	45 (1.48%)	0.0158524	ko00270
38	Ether lipid metabolism	17 (0.56%)	0.01740316	ko00565
39	Glucosinolate biosynthesis	15 (0.49%)	0.01936801	ko00966
**2C. KEGG pathways of DEGs in CK-L vs. S-L**				
#	**Pathway**	**Annotation (79)**	***p*-Value**	**Pathway ID**
1	**ABC transporters**	**5 (6.33%)**	**0.0003552804**	**ko02010**
2	**Cutin, suberin, and wax biosynthesis**	**3 (3.8%)**	**0.01149507**	**ko00073**
3	Regulation of autophagy	3 (3.8%)	0.01838196	ko04140
**2D. KEGG pathways of DEGs in CK-R vs. S-R**				
#	**Pathway**	**Annotation (502)**	***p*-Value**	**Pathway ID**
1	Nitrogen metabolism	20 (3.98%)	8.034565e-12	ko00910
2	Endocytosis	38 (7.57%)	9.001716e-06	ko04144
3	Phenylpropanoid biosynthesis	26 (5.18%)	3.331352e-05	ko00940
4	Ether lipid metabolism	8 (1.59%)	0.0002257619	ko00565
5	Glucosinolate biosynthesis	7 (1.39%)	0.0004888917	ko00966
6	**Biosynthesis of secondary metabolites**	**89 (17.73%)**	**0.0005704914**	**ko01110**
7	Plant-pathogen interaction	40 (7.97%)	0.004413053	ko04626
8	**Metabolic pathways**	**135 (26.89%)**	**0.004841477**	**ko01100**
9	ABC transporters	10 (1.99%)	0.005615396	ko02010
10	Arginine and proline metabolism	9 (1.79%)	0.006949714	ko00330
11	Flavonoid biosynthesis	8 (1.59%)	0.010255	ko00941
12	Indole alkaloid biosynthesis	4 (0.8%)	0.01178287	ko00901
13	Protein processing in endoplasmic reticulum	24 (4.78%)	0.01324247	ko04141
14	Plant hormone signal transduction	30 (5.98%)	0.01431411	ko04075
15	Pyruvate metabolism	12 (2.39%)	0.01621669	ko00620
16	Tryptophan metabolism	8 (1.59%)	0.01940078	ko00380

The Kyoto Encyclopedia of Genes and Genomes (KEGG) pathway analysis was performed using DEGs from four treatment groups. CK-L (CK, leaves) vs. D-L (drought, leaves), CK-R (CK, roots) vs. D-R (drought, roots), CK-L vs. S-L (salinity, leaves), CK-R vs. S-R (salinity, roots) represent DEGs under these two abiotic stresses. Bolded rows represent the most highly enriched metabolic pathways.

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
