# Peer review of "Physiological and Transcriptional Responses of Industrial Rapeseed (Brassica napus) Seedlings to Drought and Salinity Stress"

_ijms, 2019, doi:10.3390/ijms20225604_

Round 1

Reviewer 1 Report

My major concern is about RNA-seq description. It is unclear from Materials and Methods what plants were sequenced. If there is three replicates of two organs in two sets of control+experiment, it should be 24 plants, but in Methods 18 plants is mentioned. So it is not clear how many biological replicates were sequenced. It should be at least two biological replicates. Please, clarify it and perform additional RNA-seq if necessary.

Section 4.3 of Materials and Methods lacks mandatory details. Please, provide more detailed description of RNA-seq: sequencing library type (polyA extraction or rRNA depletion, strand-specific or not, etc), read length, read type.

Please provide the total number of reads for each sample sequenced.

In section 4.4 of Materials and Methods provide settings for each software.

Please, provide full lists of differential expressed genes as a supplementary tables.

There is no access link to raw RNA-seq data, please add a link to a database where the data are uploaded.

In Figure 1 the indication of significant difference between samples with letters is unclear. Lines 80-84 read as there is no difference between treatments in Figure 1F–G, and there is a difference in Figure 1H, but in Figure 1F–G the indicating letters are ‘a’, ‘b’ and ‘ab’ and in Figure 1H - only ‘a’ and ‘b’. Please, clarify it.

Figure 1 is also misleading because of different scales at each graph, so it is hard to see that there is a difference in Figure 1F but no difference in Figure 1C or 1D.

In section 2.4 of Results it is not clear, was the GO enrichment analysis performed for DEGs or was it a simple description of DEGs GO annotation? Does Figure 4 represent only enriched categories? Also I have no idea, what the numbers at the right ("y-axis on the right"?) mean in Figure 4. Please, clarify it.

The discussion of GO analysis (lines 303-308) considers very side terms, which do not describe any particular process and so are quite useless in discussion.

What was the method of TF enrichment analysis in section 2.5 of Results? It is not described in Materials and Methods. Is it statistically significant or is it a consequence of different overall distribution of TFs in two organs, not the differential expressed TFs?

Though the differential expression of genes involved in proline metabolism together with difference in proline content under stress conditions suggest the association of proline with stress response in Brassica napus, the evidences of regulation of the process by any TF class are weak and do not even based on the analysis of overrepresentation of binding site motifs in promoters of DEGs.

Minor comments

Many references of figures and tables in the manuscript have a dot (e.g. (Figure 2.)), which does not meet journal standards. Please, correct.

Line 63: Double dot, please correct.

Line 112-116 will be more readable as table. Please, consider it.

Section 2.6: is the description of tested genes necessary? Please, consider the transfer of this information in supplements.

Lines 230-232: species names should be in italic, please correct it, and check the species throughout the manuscript.

Gene names should be in italic, please, check it throughout the manuscript.

Line 290: the identification of binding sites of TFs do not appear in the Results, only in Discussion. Please, add the relevant section in Results.

Author Response

Point 1:My major concern is about RNA-seq description. It is unclear from Materials and Methods what plants were sequenced. If there is three replicates of two organs in two sets of control+experiment, it should be 24 plants, but in Methods 18 plants is mentioned. So it is not clear how many biological replicates were sequenced. It should be at least two biological replicates. Please, clarify it and perform additional RNA-seq if necessary.

Response 1: Thanks for the advice. There were 3 sets of sequencing samples including the control (CK), salinity-treated group and drought-treated group. Every set included two organs (leaves and roots), where each organ has three replicates. Therefore, it should be 3*2*3=18 plants in total.

Point 2:Section 4.3 of Materials and Methods lacks mandatory details. Please, provide more detailed description of RNA-seq: sequencing library type (polyA extraction or rRNA depletion, strand-specific or not, etc), read length, read type.

Response2: Thanks for the advice. We have added the mandatory details accordingly.

Point 3: Please provide the total number of reads for each sample sequenced.

Response 3: Thank you, we have modified Table 1.

Point 4:In section 4.4 of Materials and Method As provide settings for each software.

Response 4: Thanks, settings for each software have been added.

Point 5:Please, provide full lists of differential expressed genes as a supplementary tables.

Response 5: Thank you. The full lists of differential expressed genes has been provided in supplementary materials (Table S1).

Point 6: There is no access link to raw RNA-seq data, please add a link to a database where the data are uploaded.

Response 6: Thank you for your guidance. The RNA-seq data was uploaded to NCBI, the accession number is PRJNA579479.

Point 7: In Figure 1 the indication of significant difference between samples with letters is unclear. Lines 80-84 read as there is no difference between treatments in Figure 1F–G, and there is a difference in Figure 1H, but in Figure 1F–G the indicating letters are ‘a’, ‘b’ and ‘ab’ and in Figure 1H - only ‘a’ and ‘b’. Please, clarify it.

Response 7: Thanks, you are right. We modified the corresponding sentence.

Point 8: Figure 1 is also misleading because of different scales at each graph, so it is hard to see that there is a difference in Figure 1F but no difference in Figure 1C or 1D.

Response 8: Sorry for the misleading in Figure 1. Since each sub-plot measured the different physiological parameters, it is hard to modify these scales. We double checked our statistic analysis, the analysis were all correct. Some of data may have big variance to cause no difference.

Point 9: In section 2.4 of Results it is not clear, was the GO enrichment analysis performed for DEGs or was it a simple description of DEGs GO annotation? Does Figure 4 represent only enriched categories? Also I have no idea, what the numbers at the right ("y-axis on the right"?) mean in Figure 4. Please, clarify it.

Response 9: Sorry for the unclear information. The GO enrichment analysis was performed for all of the DEGs, but not every DEG could get a GO annotation and sometimes one DEG could get several annotations based on WEGO software. Figure 4 represents enriched categories, and the y-axis on the right indicates the number of DEGs that get GO annotations. For example, ‘812’ out of 913 DEGs (CK-L vs. D-L) could get annotations, and ‘812’ was divided by tenfold into ‘81’ for its visualization (Ye J, Zhang Y, et al. Nucleic Acids Res. 2018 Jul 2;46(W1):W71-W75.).

Point 10:The discussion of GO analysis (lines 303-308) considers very side terms, which do not describe any particular process and so are quite useless in discussion.

Response 10: Thanks for your suggestion. We already deleted those sentences.

Point 11:What was the method of TF enrichment analysis in section 2.5 of Results? It is not described in Materials and Methods. Is it statistically significant or is it a consequence of different overall distribution of TFs in two organs, not the differential expressed TFs?

Response 11: Thank you for your advice. We have added the sentences in Materials and Methods.

Point 12:Though the differential expression of genes involved in proline metabolism together with difference in proline content under stress conditions suggest the association of proline with stress response in Brassica napus, the evidences of regulation of the process by any TF class are weak and do not even based on the analysis of overrepresentation of binding site motifs in promoters of DEGs.

Response 12: Thank you for your advice. We have modified the sentences in discussion (please see Supplementary Figure 3).

Point 13: Minor comments

Many references of figures and tables in the manuscript have a dot (e.g. (Figure 2.)), which does not meet journal standards. Please, correct.

Response 13: Thank you so much. We already modified those.

Line 63: Double dot, please correct.

Thank you so much. We already modified those.

Line 112-116 will be more readable as table. Please, consider it.

Thanks you so much. We already added the information into Table 1.

Section 2.6: is the description of tested genes necessary? Please, consider the transfer of this information in supplements.

Thank you so much. I already transfer those information into Supplementary Table 1.

Lines 230-232: species names should be in italic, please correct it, and check the species throughout the manuscript.

Thank you so much. We checked the manuscript now.

Line 290: the identification of binding sites of TFs do not appear in the Results, only in Discussion. Please, add the relevant section in Results.

Thank you so much. We added some sentence in Results now.

Reviewer 2 Report

The article by Wang, et al. describes a detailed study of B. napus responses to abiotic stresses at both the transcriptional and, to some degree, physiological levels.  They made a special effort to detail changes in the pathways for proline synthesis and catabolism, and an unusual effort to display where some of the transcription factors they found might have been binding 5’ to several genes of the proline pathway.

The paper is quite well written, and nearly free of typographical errors.  I believe it would be a solid contribution to researchers of osmotic stress responses in plants.

The following are a few additional comments:

Line 63: silicon, not sillicon;

Line 70:  I am not sure they can conclude that “proline metabolism plays a more important role …than ROS removal”.  All of their studies are observational and don’t address which defense processes are rate-limiting or essential, and which are incidental.  Moreover, all of the physiological responses, and many of the transcriptional responses, are by factors of 2.  It is not clear to me that a 2-fold increase in proline during a PEG response (Fig. 1) tells us more, or is more functionally significant, than the 2-fold increase in H2O2 during salt stress.  (it seems possible that the increase in H2O2 might just mean that the plant has a lot of SOD waiting to deal with superoxide radicals and so, at this time, doesn’t need to make more SOD or even more catalase to eliminate the H2O2.)  I suggest they eliminate these kinds of judgements concerning their observations.

Figure 3:  This is a very personal opinion, but I for one can’t get much out of the heatmap except for general trends.  If any figure needs to be shunted to the supplements, I would vote for this one.  Wherever it goes, could the authors please explain what the dendritic tree to the left of the figure is, and what we should learn from it?

Table 2:  I am probably also alone in wondering what to conclude from KEGG analysis.  Perhaps the authors could clarify how the category “metabolic pathways” differs in components from the category “biosynthesis of secondary metabolites”?

Figure 5:  this study of transcription factors is interesting, and should be kept, but I feel the authors should modify some of their language to acknowledge that they are looking at families not individual TFs.  There are many WRKY proteins.  They, as well as MYBs and EREBPs, do widely different things. Writing about changes in them as a class doesn’t tell us how many members of each family were reacting to the treatments, nor which factors were.  Talking about them as if they are interchangeable seems to obscure the story.  I did like their search for TF binding sites in the 5’ region of the genes, but 1st, this doesn’t tell us which are responsible for the effects observed in this paper, nor 2nd, whether binding sites farther away or 3’ to the gene might have roles as well.  Now, if they did CHIP or any assay that measured occupation of those sites in control and treatment conditions, that would be spectacular.

Figure 6:  I saw that I could have looked up the fold-change for each gene in the transcriptome but would have preferred if the authors did that and then showed how those numbers agreed with the PCRs here.

Figure 6:  I confess, I was a bit disturbed that the source materials for the PCRs were the same used in the transcriptome (line 399).  This felt like double-dipping.  A far more meaningful test would have been to use independent samples to show the analysis gave predictive results for future experiments of the same kind.  However, I have seen other papers do the same thing and so don’t have strong feelings whether these authors should be asked to redo the tests.

Finally, I question whether 2-fold changes in any gene or solute are likely to have much of an affect?  Is it possible they are just biological noise (even if reproducible) as new equilibria or steady states are established?  I would have chosen more stringent conditions (at least 4-fold, or higher) unless I had evidence that there is a precedent for reporting lower values.  For me, these long lists of genes obscure the genes that are more likely to be part of the defense, or killing, of the plant. 

Author Response

Point 1:Line 63: silicon, not sillicon;

Response 1: Sorry for this mistake. We have corrected this error.

Point 2: Line 70: I am not sure they can conclude that “proline metabolism plays a more important role …than ROS removal”. All of their studies are observational and don’t address which defense processes are rate-limiting or essential, and which are incidental. Moreover, all of the physiological responses, and many of the transcriptional responses, are by factors of 2. It is not clear to me that a 2-fold increase in proline during a PEG response (Fig. 1) tells us more, or is more functionally significant, than the 2-fold increase in H2O2 during salt stress. (it seems possible that the increase in H2O2 might just mean that the plant has a lot of SOD waiting to deal with superoxide radicals and so, at this time, doesn’t need to make more SOD or even more catalase to eliminate the H2O2.) I suggest they eliminate these kinds of judgements concerning their observations.

Response 2: Thanks for your advice. We have modified the sentence accordingly.

Point 3: Figure 3: This is a very personal opinion, but I for one can’t get much out of the heatmap except for general trends. If any figure needs to be shunted to the supplements, I would vote for this one. Wherever it goes, could the authors please explain what the dendritic tree to the left of the figure is, and what we should learn from it?

Response 3: Thanks for your advice, we have transferred it to the supplements.

Point 4: Table 2: I am probably also alone in wondering what to conclude from KEGG analysis. Perhaps the authors could clarify how the category “metabolic pathways” differs in components from the category “biosynthesis of secondary metabolites”?

Response 4: Thank you for your question. According to KEGG database, 7 major pathway are described: 1. Metabolism; 2. Genetic Information Processing; 3. Environmental Information Processing; 4. Cellular Processes; 5. Organismal Systems; 6. Human Diseases; 7. Drug Development (please see the following link for details. https://www.genome.jp/kegg/pathway.html#metabolism). In metabolism, global and overview maps, carbohydrate metabolism and other 10 pathways were listed. Both metabolic pathways and biosynthesis of secondary metabolites were from global and overview maps section in metabolism. After we checked those two pathways’ description in KEGG database, biosynthesis of secondary metabolites looks like to be included in metabolic pathways. So we then carefully checked the exact DEGs which listed in biosynthesis of secondary metabolites section and found that some DEGs could be classified into different pathways. So based on our understanding, KEGG analysis could give your some general ideas about which pathway were involved. Those pathway itself may overlap in KEGG database to some extent. This similar KEGG analysis could be also found in some literatures (Shaofeng Li, Chengming Fan et.al., Effects of drought and salt-stresses on gene expression in Caragana korshinskii seedlings revealed by RNA-seq, BMC Genomics ,2016: 17:200).

Point 5:Figure 5: this study of transcription factors is interesting, and should be kept, but I feel the authors should modify some of their language to acknowledge that they are looking at families not individual TFs. There are many WRKY proteins. They, as well as MYBs and EREBPs, do widely different things. Writing about changes in them as a class doesn’t tell us how many members of each family were reacting to the treatments, nor which factors were. Talking about them as if they are interchangeable seems to obscure the story. I did like their search for TF binding sites in the 5’ region of the genes, but 1st, this doesn’t tell us which are responsible for the effects observed in this paper, nor 2nd, whether binding sites farther away or 3’ to the gene might have roles as well. Now, if they did CHIP or any assay that measured occupation of those sites in control and treatment conditions, that would be spectacular.

Response 5: Thanks for your advice, we have modified languages accordingly. We also analyst our MYB and AP2/EREBP transcription factor families and identified the tip hits with known transcription factors. CHIP could provide direct downstream targets of transcriptional factors and definitely worth further investigations.

Point 6::Figure 6:  I saw that I could have looked up the fold-change for each gene in the transcriptome but would have preferred if the authors did that and then showed how those numbers agreed with the PCRs here.

Response 6: Thank you for your suggestions. We have added the fold-change together.

Point 7:Figure 6: I confess, I was a bit disturbed that the source materials for the PCRs were the same used in the transcriptome (line 399). This felt like double-dipping. A far more meaningful test would have been to use independent samples to show the analysis gave predictive results for future experiments of the same kind. However, I have seen other papers do the same thing and so don’t have strong feelings whether these authors should be asked to redo the tests.

Response 7: Thank you for your advice and guidance. Right now, qPCR is commonly used to further verify the reliability of the sequencing data which means the same extracted RNA must to be used as templates (Loraine AE, Mccormick S, Estrada A, et al. RNA-Seq of Arabidopsis Pollen Uncovers Novel Transcription and Alternative Splicing. Plant physiology, 2013, 162(2):1092-1109; Vaneechoutte D, Estrada A R , Lin Y C , et al; Ma L, Coulter J, Liu L, et al. Transcriptome Analysis Reveals Key Cold-Stress-Responsive Genes in Winter Rapeseed (Brassica rapa L.). International Journal of Molecular Sciences, 2019, 20(5).). The main purpose is that transcriptome generally had huge data, those data should be validated using the other techniques. However, we thought your advice should be widely considered later. For example, when some candidate genes were chosen and needed to be further characterized based on trancriptome, independent samples under the same treatments should be used to avoid contingency.

Point 8: Finally, I question whether 2-fold changes in any gene or solute are likely to have much of an affect? Is it possible they are just biological noise (even if reproducible) as new equilibria or steady states are established? I would have chosen more stringent conditions (at least 4-fold, or higher) unless I had evidence that there is a precedent for reporting lower values. For me, these long lists of genes obscure the genes that are more likely to be part of the defense, or killing, of the plant.

Response 8: Thank you for your questions. DEGs are usually identified based on the parameters of an adjusted fold change of ≥ 2.00 and P-value ≤ 0.05 at the same time just as we did in this manuscript (Ma L , Coulter J , Liu L , et al. Transcriptome Analysis Reveals Key Cold-Stress-Responsive Genes in Winter Rapeseed (Brassica rapa L.). International Journal of Molecular Sciences, 2019, 20(5);Li S , Fan C , Li Y , et al. Effects of drought and salt-stresses on gene expression in Caragana korshinskii seedlings revealed by RNA-seq. BMC Genomics, 2016, 17(1):200.). But as you suggested, higher threshold could minimize those noise. In this manuscript, we redid the DEGs table using 4 fold as the supplementary table for the extra reference. Number of DEGs have been greatly decreased (Table S1, fold change>4).